# Design, Synthesis and Biological Evaluation of α-Synuclein Proteolysis-Targeting Chimeras

**DOI:** 10.3390/molecules28114458

**Published:** 2023-05-31

**Authors:** Tianzhi Wen, Jian Chen, Wenqian Zhang, Jiyan Pang

**Affiliations:** School of Chemistry, Sun Yat-sen University, Guangzhou 510006, China

**Keywords:** α-synuclein, protein aggregates, PROTAC (proteolysis-targeting chimera)

## Abstract

α-Synuclein aggregation under pathological conditions is one of the causes of related neurodegenerative diseases. PROTACs (proteolysis targeting chimeras) are bifunctional small molecules that induce a post-translational erasure of proteins via the ubiquitination of target proteins by E3 ubiquitin ligase and subsequent proteasomal degradation. However, few research studies have been conducted for targeted protein degradation of α-synuclein aggregates. In this article, we have designed and synthesized a series of small-molecule degraders **1**–**9** based on a known α-synuclein aggregation inhibitor sery384. In silico docking studies of sery384 with α-synuclein aggregates were accomplished to ensure that the compounds bound to α-synuclein aggregates specifically. The protein level of α-synuclein aggregates was determined to evaluate the degradation efficiency of PROTAC molecules on α-synuclein aggregates in vitro. The results show that compound **5** had the most significant degradation effect, with DC_50_ of 5.049 μM, and could induce the degradation of α-synuclein aggregates in a time- and dose-dependent manner in vitro. Furthermore, compound **5** could inhibit the elevation of the ROS level caused by overexpression and aggregation of α-synuclein and protect H293T cells from α-synuclein toxicity. Conclusively, our results provide a new class of small-molecule degraders and an experimental basis for the treatment of α-synuclein related neurodegenerative diseases.

## 1. Introduction

α-Synuclein is a soluble protein expressed in the presynaptic and perinuclear regions of the central nervous system. It is the main component of Lewy bodies and is closely related to the pathogenesis of related degenerative deceases. In pathological conditions, synaptic nuclear proteins are inclined to aggregate to form insoluble fibril deposits, which eventually lead to the death of nerve cells [1]. The general methods for targeting synucleinopathy are summarized as reducing the level of α-synuclein and inhibiting the aggregation of α-synuclein. There are preclinical studies in which research groups have developed some experimental tactics based on these strategies, and the availability was tested on pathological models [2,3,4,5]. However, these studies had limitations, such as inefficient delivery to the target cells in vivo, off-target effects and low utilization. Therefore, novel approaches need to be developed to specifically wipe out α-synuclein by force.

In recent years, researchers have developed a series of new strategies for targeted protein degradation with specificity and higher efficiency, and proteolysis-targeting chimera (PROTAC) has emerged. PROTACs are bifunctional molecules that can simultaneously bind the E3 ubiquitin ligase and the protein of interest (POI). By “hijacking” the ubiquitin–proteasome system (UPS), lysines exposed to the targeted proteins are ubiquitinated by the E3 ubiquitin enzyme complex, and then the POI is degraded by the proteasome. Unlike traditional methods, PROTACs aim to erase the targeted proteins at the post-translational level rather than inhibit them. This targeted protein degradation technology has a great potential to target “undruggable” proteins, which comprise approximately 85% of all human proteins [6,7,8].

For many years, researchers have been attempting to use the PROTAC technology to reduce the level of pathological protein aggregates in neurodegenerative diseases [9]. In 2016, Chu et al. used peptide-based PROTAC compounds to target tau protein, which is an important pathological protein of Alzheimer’s disease (AD), for protein degradation [10]. In 2017, Tomoshige et al. designed a series of PROTAC compounds that have a Huntingtin (HTT)-binding motif and a cellular inhibitor of the apoptosis protein 1 (cIAP1) E3 ligase binding motif [11].

PROTAC appears to be a promising method for targeted protein degradation of α-synuclein aggregates, yet little research has been conducted. In this study, we designed and synthesized a series of new bifunctional molecules based on the α-synuclein aggregation inhibitor (sery384) [12] and common ligands of E3 ligases (Cereblon, von Hippel–Lindau, cIAP1). Linkers with different numbers of units of ethylene glycol (for example, diethylene glycol or triethylene glycol) and alkyl linkers (6 carbons) were applied for the design of PROTACs of various lengths. The degradation and neuroprotective effects were evaluated by biological experiments on a cellular model in which α-synuclein overexpressed H293T cells (human embryo kidney cells) were transfected with preformed fibril (PFF) for aggregation promotion.

## 2. Results and Discussion

### 2.1. Design of Small-Molecule PROTACs for α-Synuclein Aggregates

A bifunctional molecule that achieves chemical-induced degradation of α-synuclein aggregates was composed of a binder of α-synuclein fibril, a linker and an E3 ligase ligand (Figure 1a). A di-phenyl-pyrazole derivative 3-(4-Aminophenyl)-5-(3-bromophenyl)-1H-pyrazole named sery384 [12] is a known aggregation inhibitor of α-synuclein. In vitro, sery384 could intercept the pathological aggregation of α-synuclein. Here, the molecular docking simulation shows that two hydrogen atoms of the pyrazole ring and the amine group of sery384 were bound to the hydroxyl group of the residue Ser42 and Thr44 of the crystal structure α-synuclein aggregates (PDB: 2N0A) [13], respectively (Figure 1b). Hence, we decided to utilize sery384 as a warhead and constructed heterobifunctional molecules. In terms of the E3 ligase ligand, bestatin, VH032 and pomalidomide, which interact with cIAP1 [14], VHL [15] and CRBN [16], respectively, were chosen to recruit the corresponding E3 ligase. As for the linkers that connected the warhead moiety and E3 ligand moiety, polyethylene glycol (PEG)-based linkers and alkyl linkers were applied to explore the effects on degradation altered by the length and hydrophilicity of small-molecule PROTACs. Accordingly, we designed the PROTAC molecules by connecting sery384 and VH032 (VHL ligand), bestatin (cIAP1 ligand) and pomalidomide (CRBN ligand) as ligands for the E3 ligases (Figure 1c).

### 2.2. Chemical Synthesis

The synthetic routes to compounds **1**–**9** are shown in Figure 1 and Figure 2, and all the new compounds were fully characterized by MS and NMR. The pyrazole ring of compound **10** was firstly protected with 3,4-dihydro-2H-pyran (DHP) to obtain compound **11**, considering it may also attack the electrophilic linker intermediates (such as **13**, **14**, **19**, **22**, **23** and **28**). Then, the nitro group on the benzene ring was reduced to an amino group by a palladium–carbon catalytic reduction to produce compound **12**. After securing sery384 analogue **12**, pre-functionalized linker analogues **13**, **14**, **19**, **22**, **23** and **28** were connected to the intermediate by nucleophilic substitution to produce **15**, **17**, **20**, **24**, **26** and **29**, respectively. All of the protective groups were then removed from the treatment with TFA (trifluoroacetic acid). Fluoro-substituted pomalidomide was reacted with sery384 analogues (**16**, **18** and **21**) by nucleophilic aromatic substitution to provide compounds **1**, **2** and **3**, respectively (Figure 1). The PROTACs **4**, **5** and **6** were synthesized by an amide condensation reaction of **16**, **18** and **21** with N-Boc bestatin at room temperature, respectively, followed by deprotecting by the treatment with TFA (Figure 1). The PROTACs **7**, **8** and **9** were synthesized by an amide condensation reaction of **25**, **27** and **30** with VH032 at room temperature, respectively (Figure 2).

### 2.3. Compounds Lowered the Level of α-Synuclein Aggregates In Vitro

We evaluated the degradation effect on α-synuclein aggregates based on the established cellular model according to the protocol (detailed in Section 3). In this model, α-synuclein overexpressed HEK293T cells (human embryo kidney cells) were transfected with preformed fibril (PFF), which served as the nuclei and accelerated the aggregation of α-synuclein in vitro. The results (Figure 2) show that most of the compounds had a potent degradation effect. Among them, the degraders based on the CRBN ligand (**1**–**3**) and the cIAP1 ligand (**4**–**6**) showed a greater level of degradation when compared with compounds composed of the VHL ligand (**7**–**9**). Compound **5** showed the greatest level of degradation and exhibited the lowest half degradation concentration (DC_50_) of 5.049 μM, and the protein level of the α-synuclein aggregates was reduced to approximately 20% of the normalized protein level. Furthermore, as shown in Figure 2, the selection of linkers in the PROTAC molecules were not significantly related to the degradation level of the compounds, although the compounds based on PEG-3 linkers seemed to have a more potent degradation effect on the α-synuclein aggregates than compounds based on PEG-2 linkers, demonstrating that increasing the length of linkers could improve the compounds’ flexibility, thereby enabling α-synuclein aggregates to interact with E3 ligases more effectively.

### 2.4. Compound **5** Induced Degradation of α-Synuclein Aggregates in a Dose- and Time- Dependent Manner

The degradation of the α-synuclein aggregates of compound **5** was examined in the cells in a dose- and time-dependent manner. For this purpose, a cellular model was constructed on H293T cells and treated with compound **5** at different concentrations (from 1.25 to 20 μM) for 48 h. The Western blot results show that the levels of α-synuclein aggregates were reduced in a dose-dependent manner. When the cells were treated with 10 μM of compound **5** in 48 h, the effect of α-synuclein degradation was the most promising (Figure 3a,b). Furthermore, the degradation dynamics were monitored after a treatment of 20 μM of **5** for 0 to 48 h at various time periods. The results show that the level of α-synuclein aggregates gradually decreased within the incubation period. This type of change became statistically significant and reached its peak after 24 h (Figure 3c,d). In conclusion, these data suggest that this PROTAC molecule can induce a dose- and time-dependent reduction in the level of α-synuclein aggregates in cells.

### 2.5. Compound **5** Rescued the Elevation of ROS Level Caused by α-Synuclein Overexpression and Aggregation

There has been evidence revealed that oxidative stress promotes the overexpression and aggregation of α-synuclein, and α-synuclein aggregates may in turn exacerbate oxidative stress, forming a vicious cycle [17,18,19,20]. We determined whether the treatment of compound **5** could rescue the elevation of the ROS level in α-synuclein overexpressing cells. A cellular model was constructed on H293T cells and treated with 20 μM compound **5** for 24 h. The cellular ROS level was assessed by DCFH-DA staining under confocal microscopy. The results and a quantitative analysis (Figure 4) show that, compared with the non-treated wild-type cells (control), the elevation of the ROS level in the model cells was alleviated by treatment with compound **5**. The results above indicate that compound **5** could inhibit the uprising of ROS induced by the aggregation of α-synuclein and protect cells from α-synuclein aggregates-associated toxicity. Additionally, a cytotoxicity assay of H293T cells with two representative compounds, **4** and **5** (24 h), was conducted, and the data are provided in the Appendix A. At the tested concentrations, **4** and **5** showed light effects (>25 μM) on cell viability. The log*P* values of **1**–**9** were preliminary predicted using Chem3D 20.0.0.41, and the data are provided in the Appendix A. The predicted results show that the compounds theoretically had good potential in penetrating the blood–brain barrier, but had low oral bioavailability.

## 3. Materials and Methods

### 3.1. General Information

All the reagents and solvents were purchased from commercial suppliers and used without further purification. Column chromatography was performed on silica gel (200–300 mesh, Qingdao Haiyang Chemical Co. Ltd., Qingdao, China). The NMR data were recorded in methanol-*d4* or chloroform-*d*, using TMS as an internal reference on a Bruker Avance NMR spectrometer (^1^H, 400/600 MHz; ^13^C, 100/150 MHz, Bruker Corporation, Billerica, MA, USA). The LRMS data were recorded on a Thermo LTQ XL ion trap mass spectrometer. The HRMS data were measured using Thermo Q Exactive Orbitrap High Resolution mass spectrometry (Thermo Fisher Scientific Inc., Waltham, MA, USA).

### 3.2. Synthetic Procedures

#### 3.2.1. General Methods

##### General Method A

The corresponding amine (1 eq) and the corresponding acid (1 eq) were dissolved in 3 mL of DMF, and then DIPEA (4 eq) and HATU (1.25 eq) were added. The solution was stirred at room temperature for 0.5 h, quenched with water after the reaction and extracted with ethyl acetate. The crude product was purified by column chromatography and dried under a reduced-pressure vacuum to obtain the corresponding amide.

##### General Method B

The corresponding compound was dissolved in 2 mL of dichloromethane, and then 2 mL of Trifluoroacetic acid was added. The solution was stirred at room temperature for 0.5 h and vacuumed under reduced pressure to obtain the corresponding product. The crude product needed no further purification.

##### General Method C

The corresponding amine (1 eq) was dissolved in ethyl acetate, and then triethylamine (2 eq) and Di-tert-butyl dicarbonate (2 eq) were added. The solution was refluxed overnight. After the reaction, the system was allowed to cool to room temperature and the solvent was removed from the reduced pressure. The crude product was purified by column chromatography to obtain the corresponding amide.

##### General Method D

The corresponding alcohol (1 eq) was dissolved in dichloromethane, and then triethylamine (2 eq), Tosyl chloride (2 eq) and 4-Dimethylaminopyridine (30 mg) were added successively. The solution was stirred at room temperature for 4 h. After the reaction, the solvent was removed from the reduced pressure, and the crude product was purified by column chromatography to obtain the corresponding sulfonic ester.

##### General Method E

The corresponding amine (1 eq) and the corresponding sulfonic ester (1 eq) were dissolved in 3 mL of DMF, and then potassium carbonate (2 eq) and potassium iodide (2 eq) were added. The solution was heated to 90 °C and refluxed for 24 h. After the reaction, the system was allowed to cool to room temperature, diluted with water, and extracted with ethyl acetate. The organic layer was combined and washed with water and brine, respectively. The crude product was purified by column chromatography to obtain the corresponding product.

#### 3.2.2. Synthesis of Sery384 Analogue **12**

p-Toluenesulfonic acid (15 mg) and 3,4-Dihydro-2H-pyran (2 eq, 183 μL) were added to a solution of compound **10** (1 eq, 344 mg) in 10 mL of anhydrous tetrahydrofuran. The solution was stirred at room temperature overnight. After the reaction, the solvent was removed from the reduced pressure. The crude product was purified by column chromatography (petroleum ether/ethyl acetate, 4:1) and dried in a reduced-pressure vacuum to obtain compound **11** as a yellow solid. Yield: 66%.

Compound **11** (428 mg) and 25 mg of 10% palladium carbon was suspended in 15 mL of anhydrous tetrahydrofuran. The solution was purged and refilled with hydrogen 3 times. The solution was stirred at room temperature for 1–2 h. After the reaction, the insoluble matter was filtered out with Celite diatomite, the filtrate was collected, and the solvent was removed from the reduced pressure. The crude product was purified by column chromatography (petroleum ether/ethyl acetate, 2:1), and dried in a reduced-pressure vacuum to obtain compound **12** as a yellow solid. Yield: 52%.

#### 3.2.3. Synthesis of N-Boc Bestatin

A total of 1 mL 2N sodium hydroxide aqueous solution and Di-tert-butyl dicarbonate (2 eq, 459 μL) were added to a suspension of bestatin (1 eq, 308 mg) in 25 mL of acetone dropwise while being stirred in an ice bath. After the addition, we removed the ice bath, and the mixture was allowed to warm to room temperature. After reacting overnight, the crude product was concentrated under reduced pressure, diluted with water, and extracted with ethyl acetate. The organic layer was combined and washed with a 10% citric acid aqueous solution and brine and purified by column chromatography (petroleum ether/ethyl acetate, 1:1) to obtain compound N-Boc bestatin as a white solid. Yield: 80%.

#### 3.2.4. Synthesis of Sery384 Analogues **16**, **18**, **21**, **25**, **27** and **30**

Compound **15** was synthesized via general method E. The crude product was purified by column chromatography (petroleum ether/ethyl acetate, 1:1) and dried in a reduced-pressure vacuum. Yield: 23%.

Compound **16** was synthesized via general method B.

Compound **17** was synthesized via general method E. The crude product was purified by column chromatography (petroleum ether/ethyl acetate, 1:2) and dried in a reduced-pressure vacuum. Yield: 20%.

Compound **18** was synthesized via general method B.

Compound **20** was synthesized via general method E. The crude product was purified by column chromatography (petroleum ether/ethyl acetate, 3:1) and dried in a reduced-pressure vacuum. Yield: 17%.

Compound **21** was synthesized via general method B.

Compound **24** was synthesized via general method E. The crude product was purified by column chromatography (petroleum ether/ethyl acetate, 3:1) and dried in a reduced-pressure vacuum. Yield: 19%.

Compound **25** was synthesized via general method B.

Compound **26** was synthesized via general method E. The crude product was purified by column chromatography (petroleum ether/ethyl acetate, 2:1) and dried in a reduced-pressure vacuum. Yield: 20%.

Compound **27** was synthesized via general method B.

Compound **29** was synthesized via general method E. The crude product was purified by column chromatography (petroleum ether/ethyl acetate, 3:1) and dried in a reduced-pressure vacuum. Yield: 20%.

Compound **30** was synthesized via general method B.

#### 3.2.5. Synthesis of PROTAC Molecules **1**, **2** and **3**

DIPEA (4 eq, 66μL) was added to a solution of compound **16** (1 eq, 44.5 mg) and 2-(2,6-dioxopiperidin-3-yl)-4-fluoroisoindoline-1,3-dione (1 eq, 27.6 mg) in 3 mL of DMF. The solution was heated to 90 °C and refluxed for 24 h. After the reaction, the system was allowed to cool to room temperature, diluted with water, and extracted with ethyl acetate. The organic layer was combined and washed with water and brine, respectively. The obtained crude product was purified by column chromatography (dichloromethane/methanol, 25:1) to obtain compound **1** as a yellow-green solid. Yield: 10%. ^1^H NMR (400 MHz, Chloroform-*d*) *δ*/ppm: 2.58–2.92 (m, 4H); 3.34 (dt, *J* = 12.89 Hz, *J* = 5.39 Hz, 2H); 3.49 (m, 2H); 3.60–3.80 (m, 8H); 4.90 (dd, *J* = 5.19 Hz, *J* = 11.82 Hz, 1H); 5.38 (m, 1H); 6.54 (t, *J* = 5.90 Hz, 1H); 6.62 (dd, *J* = 8.26 Hz, *J* = 2.12 Hz, 1H); 6.67 (s, 1H); 6.71 (m, 1H); 6.96 (m, 1H); 7.10 (m, 1H); 7.31 (m, 1H); 7.47 (m, 4H); 7.75 (d, *J* = 8.03 Hz, 1H); 7.97 (s, 1H). ^13^C NMR (100 MHz, Chloroform-*d*) *δ*/ppm: 22.69; 29.48; 61.71; 68.79; 69.49; 70.73; 72.48; 76.70; 77.34; 98.79; 99.03; 171.40; 110.36; 111.82; 113.21; 116.80; 122.84; 124.17; 126.68; 126.75; 128.67; 129.90; 130.23; 130.70; 132.50; 136.11; 146.85; 148.64; 167.62; 168.94; 169.36. ESI MS (*m*/*z*): 701.60, 703.60 [M + H]^+^. HRMS (ESI): calcd. for C_34_H_33_BrN_6_O_6_: *m*/*z* = 701.1717, 703.1697, found: *m*/*z* = 701.1704, 703.1685 [M + H]^+^.

The synthetic method of compound **2** is in accordance with the route of compound **1**. The obtained crude product was purified by column chromatography (dichloromethane/methanol, 25:1) to give a yellow-green solid. Yield: 9%. ^1^H NMR (400 MHz, Chloroform-*d*) *δ*/ppm: 2.65–2.92 (m, 4H); 3.33 (t, *J* = 5.71 Hz, 2H); 3.48 (q, *J* = 5.71 Hz, 2H); 3.60–3.80 (m, 12H); 4.96 (dd, *J* = 5.27 Hz, *J* = 12.25 Hz, 1H); 5.37 (m, 1H); 6.50 (t, *J* = 5.66 Hz, 1H); 6.66 (s, 1H); 6.69 (d, *J* = 1.54 Hz, 2H); 6.92 (d, *J* = 7.97 Hz, 2H); 7.11 (d, *J* = 7.46 Hz, 2H); 7.30 (t, *J* = 7.20 Hz, 1H); 7.47 (m, 4H); 7.76 (d, *J* = 7.20 Hz, 1H); 7.98 (s, 1H). ^13^C NMR (100 MHz, Chloroform-*d*) *δ*/ppm: 22.82; 29.78; 43.37; 48.95; 70.32; 70.66; 70.78; 76.70; 77.02; 77.34; 98.79; 111.76; 113.23; 115.92; 116.82; 122.85; 123.51; 124.07; 124.17; 126.66; 128.67; 129.90; 130.24; 130.72; 136.06; 146.85; 168.93; 169.42; 171.30. ESI MS (*m*/*z*): 745.52, 747.52 [M + H]^+^. HRMS (ESI): calcd. for C_36_H_37_BrN_6_O_7_: *m*/*z* = 745.1979, 747.1959, found: *m*/*z* = 745.1969, 747.1949 [M + H]^+^.

The synthetic method of compound **3** is in accordance with the route of compound **1**. The obtained crude product was purified by column chromatography (dichloromethane/methanol, 25:1) to give a yellow-green solid. Yield: 7%. ^1^H NMR (400 MHz, Chloroform-*d*) *δ*/ppm: 1.29 (m, 4H); 1.58 (m, 4H); 2.66–2.96 (m, 4H); 3.19 (t, *J* = 6.59 Hz, 2H); 3.31 (q, *J* = 5.86 Hz, 2H); 4.95 (dd, *J* = 5.86 Hz, *J*= 12.81 Hz, 1H); 5.32 (s, 1H); 6.27 (t, *J* = 6.67 Hz, 1H); 6.66 (m, 2H); 6.71 (s, 1H); 6.76 (m, 1H); 6.91 (d, *J* = 7.92 Hz, 1H); 7.11 (m, 1H); 7.31 (m, 1H); 7.47 (m, 4H); 7.76 (t, *J* = 6.25 Hz, 1H); 7.98 (s, 1H); 8.65 (s, 1H). ^13^C NMR (100 MHz, Chloroform-*d*) *δ*/ppm: 22.70; 26.65; 26.76; 29.72; 31.63; 31.94; 43.52; 48.94; 53.43; 98.92; 109.97; 111.52; 112.86; 115.28; 116.65; 118.60; 122.87; 124.19; 125.65; 126.76; 126.86; 128.70; 128.86; 130.27; 130.83; 132.53; 136.16; 147.00; 148.73; 167.63; 168.65; 169.58; 171.18. ESI MS (*m*/*z*): 669.60, 671.60 [M + H]^+^. HRMS (ESI): calcd. for C_34_H_33_BrN_6_O_4_: *m*/*z* = 669.1819, 671.1799, found: *m*/*z* = 669.1805, 671.1786 [M + H]^+^.

#### 3.2.6. Synthesis of PROTAC Molecules **4**, **5** and **6**

Compound **31** was synthesized via general method A. The crude product was purified by column chromatography (dichloromethane/methanol, 100:3) and dried in a reduced-pressure vacuum. Yield: 16%.

Compound **4** was synthesized via general method B. ^1^H NMR (400 MHz, Methanol-*d4*) *δ*/ppm: 0.96 (m, 6H); 1.64 (m, 5H); 2.04 (q, *J* = 5.86 Hz, 1H); 2.21 (t, *J* = 6.84 Hz, 1H); 2.92 (m, 1H); 3.11 (dd, *J* = 8.04 Hz, *J* = 13.91 Hz, 1H); 3.38–3.82 (m, 13H); 4.13 (d, *J* = 3.00 Hz, 1H); 4.39 (dd, *J* = 6.26 Hz, *J* = 8.61 Hz, 1H); 5.36 (m, 1H); 6.91 (m, 3H); 7.34 (m, 6H); 7.51 (dd, *J* = 8.75 Hz, *J* = 15.54 Hz, 1H); 7.65 (d, *J* = 8.26 Hz, 2H); 7.79 (dt, *J* = 1.47 Hz, *J* = 7.77 Hz, 1H); 8.01 (t, *J* = 1.88 Hz, 1H). ^13^C NMR (100 MHz, Methanol-*d4*) *δ*/ppm: 13.05; 20.95; 21.83; 22.34; 24.59; 25.53; 26.71; 28.93; 29.21; 29.43; 31.67; 35.10; 39.02; 40.64; 44.34; 52.09; 54.78; 68.40; 68.49; 69.01; 69.88; 70.05; 98.65; 114.47; 122.44; 124.05; 126.51; 127.18; 128.09; 129.45; 130.24; 130.48; 134.44; 135.28; 171.85; 173.20. ESI MS (*m*/*z*): 735.66, 737.66 [M + H]^+^. HRMS (ESI): calcd. for C_37_H_47_BrN_6_O_5_: *m*/*z* = 735.2864, 737.2843, found: *m*/*z* = 735.2850, 737.2831 [M + H]^+^.

Compound **32** was synthesized via general method A. The crude product was purified by column chromatography (dichloromethane/methanol, 100:3) and dried in a reduced-pressure vacuum. Yield: 15%.

Compound **5** was synthesized via general method B. ^1^H NMR (400 MHz, Methanol-*d4*) *δ*/ppm: 0.96 (m, 6H); 1.62 (m, 5H); 2.04 (q, *J* = 6.34 Hz, 1H); 2.23 (m, 1H); 2.91 (m, 1H); 3.10 (dd, *J* = 8.04 Hz, *J* = 13.91 Hz, 1H); 3.36–3.82 (m, 17H); 4.12 (d, *J* = 3.22 Hz, 1H); 4.38 (dd, *J* = 6.33 Hz, *J* = 8.77 Hz, 1H); 5.36 (m, 1H); 6.96 (s, 1H); 7.01 (d, *J* = 8.55 Hz, 2H); 7.34 (m, 6H); 7.51 (d, *J* = 7.60 Hz, 1H); 7.71 (d, *J* = 8.07 Hz, 2H); 7.79 (d, *J* = 8.07 Hz, 1H); 8.01 (t, *J* = 1.91 Hz, 1H). ^13^C NMR (100 MHz, Methanol-*d4*) *δ*/ppm: 13.05; 20.95; 21.83; 22.34; 24.58; 25.54; 26.71; 28.93; 29.21; 29.44; 31.68; 35.09; 39.02; 40.64; 45.40; 52.07; 54.76; 67.74; 68.37; 68.98; 69.79; 69.91; 70.14; 70.17; 98.97; 115.89; 122.47; 124.06; 126.60; 127.17; 128.10; 128.71; 129.44; 130.58; 134.12; 135.29; 171.84; 173.18. ESI MS (*m*/*z*): 779.82, 781.82 [M + H]^+^. HRMS (ESI): calcd. for C_39_H_51_BrN_6_O_6_: *m*/*z* =779.3126, 781.3105, found: *m*/*z* = 779.3111, 781.3095 [M + H]^+^. 

Compound **33** was synthesized via general method A. The crude product was purified by column chromatography (dichloromethane/methanol, 25:1) and dried in a reduced-pressure vacuum. Yield: 12%.

Compound **6** was synthesized via general method B. ^1^H NMR (400 MHz, Methanol-*d4*) *δ*/ppm: 0.90 (m, 6H); 1.30–1.90 (m, 11H); 2.93 (m, 2H); 3.06–3.25 (m, 5H); 3.76 (m, 4H); 4.15 (s, 1H); 4.34 (m, 1H); 5.35 (t, *J* = 5.49 Hz, 1H); 6.99 (s, 1H); 7.09 (d, *J* = 7.60 Hz, 2H); 7.33–7.45 (m, 5H); 7.51 (d, *J* = 7.60 Hz, 1H); 7.60–7.83 (m, 5H); 8.01 (s, 1H). ^13^C NMR (100 MHz, Methanol-*d4*) *δ*/ppm: 15.87; 17.30; 20.94; 22.33; 26.70; 29.34; 29.42; 31.66; 35.08; 38.91; 40.71; 42.37; 52.20; 54.42; 54.79; 68.46; 99.16; 116.91; 122.48; 124.06; 126.67; 127.17; 128.10; 128.71; 128.96; 129.07; 129.45; 130.29; 130.61; 133.95; 135.31; 171.86; 172.98. ESI MS (*m*/*z*): 703.71, 705.71 [M + H]^+^. HRMS (ESI): calcd. for C_37_H_47_BrN_6_O_3_: *m*/*z* = 703.2965, 705.2945, found: *m*/*z* = 703.2955, 705.2935 [M + H]^+^.

#### 3.2.7. Synthesis of PROTAC Molecules **7**, **8** and **9**

Compound **7** was synthesized via general method A. The crude product was purified by column chromatography (dichloromethane/methanol, 25:1) and dried in a reduced-pressure vacuum. Yield: 14%. ^1^H NMR (400 MHz, Methanol-*d4*) *δ*/ppm: 1.05 (s, 9H); 2.08 (m, 1H); 2.17 (m, 1H); 2.46 (s, 3H); 2.57 (m, 2H); 3.60–3.94 (m, 12H); 4.36 (m, 1H); 4.54 (m, 3H); 4.68 (m, 1H); 5.36 (m, 1H); 6.74 (d, *J* = 8.46 Hz, 2H); 6.83 (s, 1H); 7.30–7.50 (m, 7H); 7.54 (d, *J* = 7.73 Hz, 1H); 7.78 (dt, *J* = 1.38 Hz, *J* = 7.89 Hz, 1H); 8.00 (t, *J* = 1.79 Hz, 1H); 8.85 (s, 1H). ^13^C NMR (100 MHz, Methanol-*d4*) *δ*/ppm: 14.42; 25.60; 25.65; 35.42; 35.94; 37.52; 42.32; 42.93; 56.62; 57.53; 59.43; 66.87; 69.34; 69.69; 69.99; 70.07; 112.62; 122.37; 124.04; 126.37; 127.56; 128.06; 128.94; 129.09; 130.08; 130.16; 130.27; 138.79; 149.14; 151.41; 170.76; 172.35; 173.04. ESI MS (*m*/*z*): 886.50, 888.50 [M + H]^+^. HRMS (ESI): calcd. for C_44_H_52_BrN_7_O_6_: *m*/*z* = 886.2955, 888.2935, found: *m*/*z* =886.2940, 888.2924 [M + H]^+^.

Compound **8** was synthesized via general method A. The crude product was purified by column chromatography (dichloromethane/methanol, 20:1) and dried in a reduced-pressure vacuum. Yield: 16%. ^1^H NMR (400 MHz, Chloroform-*d*) *δ*/ppm: 1.05 (s, 9H); 2.30–2.40 (m, 4H); 2.44 (s, 3H); 3.34 (t, *J* = 4.99 Hz, 2H); 3.67 (m, 14H); 4.18 (d, *J* = 11.45 Hz, 1H); 4.33 (dd, *J* = 5.29 Hz, *J* = 14.98 Hz, 1H); 4.58 (m, 3H); 4.87 (t, *J* = 7.93 Hz, 1H); 5.36 (m, 1H); 6.65 (s, 1H); 6.77 (d, *J* = 8.84 Hz, 2H); 6.97 (d, *J* = 8.84 Hz, 1H); 7.25 (m, 4H); 7.41 (m, 2H); 7.59 (d, *J* = 7.96 Hz, 2H); 7.65 (dt, *J* = 1.32 Hz, *J* = 7.84 Hz, 1H); 7.89 (t, *J* = 1.75 Hz, 1H); 8.66 (s, 1H). ^13^C NMR (100 MHz, Chloroform-*d*) *δ*/ppm: 14.15; 16.02; 22.71; 26.42; 29.34; 29.39; 29.72; 31.94; 35.25; 36.93; 37.38; 43.22; 43.54; 57.35; 57.94; 59.21; 67.26; 69.05; 70.18; 70.25; 70.33; 70.38; 70.58; 76.72; 77.04; 77.36; 98.17; 113.54; 118.70; 122.77; 124.03; 126.73; 128.08; 128.54; 129.19; 130.18; 130.51; 131.79; 135.47; 138.19; 145.77; 148.31; 148.92; 150.16; 171.35; 171.82; 172.04. ESI MS (*m*/*z*): 930.52, 932.52 [M + H]^+^. HRMS (ESI): calcd. for C_46_H_56_BrN_7_O_7_: *m*/*z* = 930.3218, 932.3197, found: *m*/*z* =930.3202, 932.3186 [M + H]^+^.

Compound **9** was synthesized via general method A. The crude product was purified by column chromatography (dichloromethane/methanol, 25:1) and dried in a reduced-pressure vacuum. Yield: 15%. ^1^H NMR (600 MHz, Chloroform-*d*) *δ*/ppm: 1.02 (s, 9H); 1.15–1.25 (m, 4H); 1.41–1.54 (m, 4H); 2.10 (m, 2H); 2.22 (m, 1H); 2.31–2.51 (m, 5H); 3.01 (t, *J* = 7.05 Hz, 2H); 3.69 (d, *J* = 9.35 Hz, 1H); 4.16 (d, *J* = 10.98 Hz, 1H); 4.32 (dd, *J* = 15.42 Hz, *J* = 5.14 Hz, 1H); 4.56 (m, 2H); 4.66 (d, *J* = 9.35 Hz, 1H); 4.77 (t, *J* = 7.24 Hz, 1H); 6.55 (d, *J* = 8.51 Hz, 2H); 6.65 (s, 1H); 6.87 (d, *J* = 8.51 Hz, 1H); 7.24 (m, 1H); 7.32 (m, 1H); 7.42 (m, 1H); 7.50 (dd, *J* = 17.50 Hz, *J* = 7.57 Hz, 2H); 7.68 (d, *J* = 7.57 Hz, 1H); 7.76 (m, 1H); 7.94 (s, 1H); 8.65 (s, 1H). ^13^C NMR (150 MHz, Chloroform-*d*) *δ*/ppm: 16.02; 25.39; 26.26; 26.47; 28.49; 28.74; 28.77; 29.70; 35.00; 35.18; 36.03; 36.59; 43.21; 50.60; 57.66; 59.15; 69.93; 98.69; 112.69; 118.96; 122.83; 124.16; 125.60; 126.88; 127.91; 128.05; 128.52; 128.73; 129.33; 129.45; 130.28; 130.64; 130.75; 131.65; 134.80; 138.11; 138.13; 148.35; 148.56; 148.68; 150.32; 171.31; 171.99; 174.21. ESI MS (*m*/*z*): 854.48, 856.48 [M + H]^+^. HRMS (ESI): calcd. for C_44_H_52_BrN_7_O_4_: *m*/*z* = 854.3057, 856.3037, found: *m*/*z* = 854.3041, 856.3022 [M + H]^+^.

#### 3.2.8. Docking Studies

In silico docking studies of sery384 with α-synuclein aggregates (PDB code: 2N0A) were accomplished using the AutoDock 4.2. The AutoDock program [21] was selected because it uses a genetic algorithm, of which the Lamarckian version was employed to generate the docking poses of ligands within known or predicted binding sites. Hydrogen and Gasteiger charges were added to the X-ray structure of the α-synuclein aggregates and sery384 in the docking experiments that were carried out. The grid box was set with dimensions of 126 × 126 × 126 points and 0.375 Å spacing to parcel up the X-ray crystal structure of the α-synuclein aggregates. The docking parameters were all set as default. The docking pose of the α-synuclein aggregates with compound sery384 is displayed in Figure 1, and the rendering of the image was generated by applying PyMol [22].

#### 3.2.9. Construction of Plasmid Vector

Gene encoding human α-synuclein (including 6×His tag) was cloned into pET28a (for prokaryotic expression) and pcDNA3.1 (for eukaryotic expression) plasmid vector (Invitrogen, Carlsbad, CA, USA), respectively. The obtained vector was transferred into the competent strains of *Escherichia coli* DH5α (purchased from the National Collection of Authenticated Cell Cultures, Shanghai, China), and was transformed, extracted and purified using a plasmid DNA extraction kit (Beyotime, Shanghai, China, Cat. No. D0018) according to the manufacturer’s manual.

#### 3.2.10. Preparation of α-Synuclein Pre-Formed Fiber (PFF)

The purified recombinant α-synuclein was diluted to 5 mg/mL with PBS in a micro-centrifuge tube and incubated at 37 °C and 1000 rpm for 7 days. After incubation, a visible amyloid deposit could be observed. The deposit was diluted to 100 μg/mL and was shattered by ultrasound for 60 s to obtain α-synuclein PFF (particle size: 100–1000 nm).

#### 3.2.11. Construction of Cell Model

The H293T cell line was purchased from the National Collection of Authenticated Cell Cultures (Shanghai, China). The cells were cultured in complete DMEM medium (supplemented with 10~15% fetal bovine serum (FBS), 100 U/mL penicillin and 100 μg/mL streptomycin) at 37 °C in an atmosphere of 5% CO_2_. All the reagents for the cell culture were purchased from Gibco (Grand Island, NY, USA). H293T cells of logarithmic growth phase were prepared and cultured in a 6-well plate. Then, pcDNA3.1-SNCA-6×His was transferred into the cells using Lipofectamine 2000 (Invitrogen). The cells were cultured for 12 h for α-synuclein expression. After 12 h, the PFF was transfected into the cells using Lipofectamine 2000, and the protein aggregates were formed after 12 h of culture.

After the construction of the cell model, the culture medium was discarded, and the precipitation was fully rinsed with PBS to remove any residual PFF. Then, the complete medium containing the corresponding compounds (concentration: 1.25~20 μM) was added, and the cells were cultured for another 24 h.

#### 3.2.12. Determination of Protein Level

After dosing, the cells were collected and an SDS lysate solution (Beyotime, Cat. No. P0013G) (containing a mixture of protease inhibitors (Beyotime, Cat. No. P1005)) was added. The cells were decomposed on ice for 20 min, followed by ultrasonic fragmentation. The solution was centrifuged at 4 °C and 12000 RCF for 10 min, and a buffer solution was added. The harvested protein sample was denatured at 95 °C for 10 min and loaded onto SDS-PAGE gel; afterwards, it was separated by electrophoresis and subsequently transferred to PVDF film for an immune reaction (Cat. No. #2642 α-Synuclein Antibody; Manufacturer: Cell Signaling Technology), and finally developed by chemiluminescence (JP-K600, Jiapeng, Shanghai, China). Image J software was used to quantify the developing image, and GAPDH was used as the internal reference to calculate the level of α-synuclein aggregates in each cell group.

#### 3.2.13. DCFH-DA Staining

After modeling and treatment, the H293T cells were fully rinsed with PBS and stained using DCFH-DA staining kits according to the manufacturer’s protocols (Beyotime, Cat. No. S0033S). The cells were subsequently washed, counterstained with Hoechst and finally observed using a confocal laser scanning microscope (LSM 710 NLO, Zeiss, Oberkochen, Germany). The resulting images were analyzed by Image J software.

## 4. Conclusions

We have designed and synthesized a series of new small-molecule PROTACs targeting α-synuclein aggregates, in which three types of common E3 ligands are linked with the known α-synuclein aggregation inhibitor sery384. Degradation and cell-protective effects were evaluated on α-synuclein overexpressed H293T cells that were transfected with PFF as seeding. Among the synthesized compounds, **5** induced the most significant degradation effect on intracellular α-synuclein aggregates. Furthermore, compound **5** rescued the elevation of ROS level by the mediated degradation of α-synuclein aggregates. In conclusion, our results provide a novel strategy for treating related neurodegenerative diseases and broaden the potential of targeted protein degradation technology.

## Data Availability

The data presented in this study can be found in the article or in the associated Appendix A.

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
