# Peer review of "Design, Synthesis and Biological Evaluation of α-Synuclein Proteolysis-Targeting Chimeras"

_molecules, 2023, doi:10.3390/molecules28114458_

Round 1

Reviewer 1 Report

Please see attachments

Only minor grammatical mistake are there, which must be refine

Reviewer 2 Report

This is a very interesting study that investigate potential use of PROTACs against Parkinson’s disease. The authors synthesized new compounds that act as a proteosome activators for alpha synuclein peptides and they found highly active compounds for this purpose. On the other hand, there are several points that should be explained before further publication process.

1- Need more detailed method for the “Construction of Plasmid Vector” part.

2- What is the cytotoxicity levels after molecule applications?

3- Why did they used H293T cells? neuronal cell cultures could be better like primer cultures or differentiated SHSY-5Y or NTERA cell cultures. Please explain.

4- There is no cytotoxicity or genotoxicity study to investigate biosafety of the synthesized molecules.

5- What is the Blood brain barrier passage potential of the molecules?

Reviewer 3 Report

Quality of Figure 1 is very poor. Please, redraw the Figure carefully.

Authors are encouraged to use an appropriate compound as a positive control in the experiments of protein degardation. 

In Materials and Methods section - Is sery384 analogue 12 - the literature compound or it was synthesized for the first time? It is nescesary to complete the synthesis with NMR data and reference. The same is for betstatin.

Minor editing of English language required

Round 2

Reviewer 2 Report

The manuscript can be accepted in the present form.

Reviewer 3 Report

Thank you for corrections

-